# Understanding the Associations across Fibromyalgia-Related Fatigue, Depression, Anxiety, Self-Esteem Satisfaction with Life and Physical Activity in Portuguese and Brazilian Patients: A Structural Equation Modeling Analysis

**DOI:** 10.3390/medicina58081097

**Published:** 2022-08-14

**Authors:** Marcos C. Alvarez, Maria Luiza L. Albuquerque, Henrique P. Neiva, Luis Cid, Diogo S. Teixeira, Rui Matos, Raúl Antunes, Liane Lúcio, Leandro Sant’Ana, Diogo Monteiro

**Affiliations:** 1Department of Sport Sciences, University of Beira Interior, 6201-001 Covilhã, Portugal; 2Research Center in Sport, Health and Human Development (CIDESD), Trás-os-Montes and Alto Douro University, 5000-558 Vila Real, Portugal; 3Life Quality Research Centre (CIEQV), 2400-901 Leiria, Portugal; 4Sport Sciences School of Rio Maior, Polytechnic of Santarém (ESDRM-IPSantarém), 2040-413 Rio Maior, Portugal; 5Research Center in Sport, Physical Education, Exercise and Health (CIDEFES), 1749-024 Lisbon, Portugal; 6Faculty of Physical Education and Sport, Lusófona University (ULHT/FEFD), 1749-024 Lisbon, Portugal; 7ESECS—Polytechnic of Leiria, 2411-901 Leiria, Portugal; 8Center for Innovative Care and Health Technology (ciTechCare), 2415-396 Leiria, Portugal; 9Primary Health Care Unit, 2040-349 Rio Maior, Portugal; 10Lisboa e Vale do Tejo—Regional Health Administration, 2005-143 Santarém, Portugal; 11Post Graduate Program in Physical Education, Federal University of Juiz de Fora, Juiz de Fora 36036-900, Brazil; 12Strength Training Studies and Research Laboratory, Federal University of Juiz de Fora, Juiz de Fora 36036-900, Brazil

**Keywords:** fibromyalgia, depression, anxiety, self-esteem, satisfaction with life, physical activity

## Abstract

*Background and Objectives:* Fibromyalgia are heterogeneous and differ from patient to patient; however, the most reported are general myalgia and at specific points associated with fatigue and certain psychological adversities. Physical activity can mitigate the effects of the symptoms. However, the associations between fibromyalgia-related fatigue, self-esteem, anxiety, depression, satisfaction with life and physical activity are unclear. Therefore, the aim of the present study was to understand the associations between these symptoms and whether there are differences between these associations across two distinct cultures. *Materials and Methods:* A total of 473 women aged between 28 and 75 years (M = 49.27; SD ± 8.28) completed five questionnaires about fibromyalgia-related fatigue, physical activity, anxiety, depression, self-esteem, and satisfaction with life. *Results:* Fibromyalgia-related fatigue was positively associated with depression and anxiety, depression and anxiety were negatively associated with self-esteem, self-esteem was positively associated with satisfaction with life, satisfaction with life was positively associated with physical activity and there were no differences in terms of the perceptions and associations of these variables between Portuguese and Brazilian patients. *Conclusions:* Our results showed the significant role of associations between these variables and a similarity in the perception and relationship of the variables between the two cultures.

## 1. Introduction

Fibromyalgia or fibromyalgia syndrome (FM) can be defined as a neurological, physical and chronic disorder that causes sensory changes and muscular pain [1]. Its pathogenesis is still under discussion by experts. Central nervous system hyperexcitability and neurotransmitter imbalance are considered the main causes of fibromyalgia [2,3]. FM affects between 2% and 6% of the world’s population, with middle-aged women (aged 30 to 50 years) the most affected [4,5]. A major characteristic of FM is its symptomatologic diversity, namely generalized muscle pain in specific points (called tender points), excessive fatigue (which is not relieved even with many hours of rest), loss of muscle strength, and psychological problems (i.e., sleep problems, anxiety, depression and reduced levels of satisfaction with life and self-esteem) [6,7].

With respect to the complexity of its symptomatology, FM patients report difficulties when reporting their symptoms to healthcare professionals, who often do not associate these clinical and somatic manifestations with FM or do not believe the patient’s report [8]. Diagnosis is complex and time-consuming, as there is still a shortage of tools for diagnosing FM [9]. Although widely used in the past, the tender point test is no longer accepted as an indicative finding for FM, so it has been excluded as a diagnostic criterion [9]. To confirm a positive diagnosis of FM, medical doctors perform a series of laboratory tests to exclude any other disease that may cause FM-like symptoms [9]. Then, patients are evaluated using the generalized pain index (GPI) and a symptom severity score (EGS). With the combination of the values obtained from both questionnaires, and no change in symptoms for three months, the patient is eventually diagnosed with FM [9,10]. It is very important in FM to receive and FM diagnosis as soon as possible because a diagnosis helps health professionals to prepare therapies to alleviate the symptoms in patients [11,12].

FM is a chronic condition that considerably affects the psychological components of patients [13]. A comparative study conducted by Bucourt et al. [14] among FM patients and patients with rheumatoid arthritis, spondylarthritis and Sjogren’s syndrome demonstrated that FM patients had higher levels of depression and anxiety and lower levels of self-esteem and life satisfaction. On the other hand, some studies [15,16] have shown that patients with FM present with significant differences at the functional level (because of fatigue levels) in terms of vitality, depression, anxiety, self-esteem and life satisfaction. To the best of our knowledge, the associations between fibromyalgia-related fatigue and the levels of depression, anxiety, self-esteem, satisfaction with life and physical activity remain unclear.

### 1.1. Fibromyalgia-Related Fatigue

Fatigue is known to be a natural reaction of the body to stress, be it physical or mental; however, it can also act as an indicator of changes in physical and mental health [17,18]. In individuals without any health problems, fatigue is, physiologically, a response to prolonged activities, which is easily alleviated after rest. However, patients with FM characterize their fatigue as excessively affecting physical, mental and cognitive levels, usually not improving, even with many hours of sleep or rest, directly interfering with their work performance or daily tasks and possibly contributing to the acceptance of less active behavior [17].

Patients with FM report that fatigue is the main symptom that directly affects their overall quality of life, as well as levels of anxiety, depression, life satisfaction and self-esteem [19,20]. Although fatigue is negatively related to various aspects of health and psychological components, there is still a dearth of information with respect to measurement of fatigue in scientific studies and clinical practice [21,22].

### 1.2. The Association between Fibromyalgia-Related Fatigue and Psychological Aspects

Due to its complexity, patients with FM tends to be more vulnerable to psychological conditions (e.g., higher levels of anxiety and depression, low self-esteem and lower levels of life satisfaction) [23]. Some studies [24,25] have shown that when compared, FM patients demonstrated higher levels of anxiety and depression than patients with other chronic diseases (i.e., rheumatoid arthritis and osteoarthritis). This increase in depressive and anxious conditions affects the social and emotional functioning of FM patients, causing them to avoid socializing, work responsibilities and any type of physical activity, causing a cycle of events that culminates in worsening of the depressive and anxious symptoms and thus reporting lower levels of self-esteem and satisfaction with life, generating a vicious circle [26,27].

Self-esteem is related to self-confidence and expectation of self-efficacy; some studies [28,29,30,31] have demonstrated that patients with FM have considerably lower levels of self-esteem in comparison to healthy individuals or those with other chronic diseases. This decrease in self-esteem levels causes a cognitive decline, directly affecting the levels of satisfaction with life, physical activity and fatigue in patients with FM [32].

Satisfaction with life is one of the dimensions associated with quality of life, which is part of the cognitive component of subjective wellbeing, which is increasingly recognized as an important health parameter; lower levels of satisfaction with life may be related to the development or worsening of chronic diseases [33,34,35] In patients with FM, previous studies [36,37,38] have concluded that higher levels of life satisfaction are associated with improved adaptation to FM (i.e., reduced levels of fatigue, anxiety, depression and disease severity).

Therefore, therapies and interventions in patients with FM should act to improve psychological components, as some studies [39,40] revealed that such interventions significantly improve the levels of psychological components, causing an improvement in patients coping with FM.

### 1.3. Physical Activity and Psychological Aspects in FM

Physical activity as a therapy for FM is highly recommended because in addition to being a low-cost intervention, it can improve physical and psychological components (increasing the activity of certain neurotransmitters, such as endorphins and serotonin) and promote a sense of satisfaction and self-efficacy [41,42].

Health professionals should always take into account the limitations of each patient and thus evaluate those of physical fitness and fatigue related to fibromyalgia so that it is possible to understand and plan the activities to be performed by patients, minimizing any adversities arising from the training program [42,43]. Patients with FM are recommended to perform physical activities of low to moderate intensity at a frequency of two to three times per week [42,43].

Patients with FM demonstrate a behavior of rejection to perform physical activities in order to preserve stressor effects, whether physical and/or psychological, which can negatively affect their physical and psychological symptomatology [44]. Some studies [45,46,47,48,49] have demonstrated that FM patients with higher levels of activity and physical fitness demonstrated lower levels of depression and anxiety.

### 1.4. Differences between Portugal and Brazil in Fibromyalgia

Despite sharing a certain similarity in their language, Portuguese and Brazilian patients have some differences in their FM-related symptoms. In Portugal, it is estimated that 2.1% of the population has a positive diagnosis for FM, with a ratio of six women for every man with a positive diagnosis [50,51]. Branco et al. [52] that Portugal, when compared with other Western European countries (i.e., Germany, Spain, Italy and France) has the highest number of patients with a positive diagnosis of FM. In Brazil, 2.5% of the population has a positive diagnosis of FM, with a ratio of 1:5 between men and women, respectively [53]. A study by Duenas et al. showed that one of the leading problems faced by Brazilian patients is difficulty in accessing treatment, care and even a diagnosis due to social disparity and the difficulty in finding health centers due to distance [54].

There is still a lack of studies analyzing and exploring these cross-cultural differences at the symptomatologic level of FM, as some studies [55,56,57] have shown that socioeconomic, demographic, climatic and even cultural reasons can alter the perception of symptoms by FM patients, causing an improvement or worsening of FM symptom levels.

Therefore, the aim of the present study is to analyze the associations between fibromyalgia-related fatigue, depression, anxiety, self-esteem, satisfaction with life and physical activity and to analyze the invariance of these relationships between Brazilian and Portuguese patients. Specifically, we explore the relationships between these psychological components. We hypothesized that (a) fibromyalgia-related fatigue is positively associated with levels of depression and anxiety and negatively related to levels of self-esteem, satisfaction with life and physical activity [58]; (b) levels of depression and anxiety are negatively associated with levels of self-esteem [38]; (c) self-esteem is positively associated with life satisfaction [23]; (d) levels of life satisfaction are positively associated with levels of physical activity [37]; and (e) there are possible differences between Portuguese and Brazilian patients in terms of the associations between the study variables, which is supported by previous studies [55,56] showing differences between cultures.

## 2. Materials and Methods

### 2.1. Study Design and Participants

Two independent samples of Portuguese (sample 1) and Brazilian (sample 2) women with a positive diagnosis of FM participated in present cross-sectional study. First, the participants were asked about their age; nationality; time since diagnosis; and levels of fatigue, depression, anxiety, self-esteem, life satisfaction and physical activity. Sample 1 comprised 222 Portuguese patients aged between 28 and 75 years (M = 49.53; SD ± 8.80). Portuguese patients were diagnosed with FM at a mean age of 9.29 ± 7.71 years. Sample 2 comprised 251 Brazilian patients with FM, aged between 27 and 75 years (M = 49.03; SD ± 8.86). These Brazilian patients had a mean time of diagnosis of 10.05 ± 7.69 years. The required sample size was determined using the Daniel Sopper online calculator [59], and the following input parameters were considered: anticipated effect size (0.3), statistical power level (0.80), number of latent variables (7) and number of observed variables (37) [60]. Therefore, the minimum sample size to detect an effect was 170, the minimum sample size for the model structure was 119 and the recommended minimum sample size was 170, which was respected in the present study for each sample.

### 2.2. Procedure: Data Collection

The Ethics and Science Committee of the University of Beira Interior, Covilhã, Portugal (UBI) granted approval for this study (reference number CE-UBIPj- 2021-038). The present study was conducted according to the standards defined by the Declaration of Helsinki [61].

For data collection, two institutions related to FM (National Association Against Fibromyalgia and Chronic Fatigue Syndrome, Lisbon, Portugal (MYOS) and the National Association of Fibromyalgia, São Paulo, Brazil (ABRAFIBRO)) were contacted, and the objectives of the study were explained. With the approval, these institutes referred some patients who voluntarily chose to participate in the study. These patients were contacted by the authors, and the purpose and objectives of the study were explained, and information was provided, with details about the data collection process. After contact and clarification, each participant filled out the informed consent, where it was explained that the individuals would participate in this study voluntarily and anonymously. The time to answer the questionnaires was approximately 35 min. MYOS is a Portuguese institution that helps people with FM and chronic fatigue syndrome and is composed of about 7000 patients with a positive diagnosis of FM, 500 of whom were contacted by the researchers, resulting in 227 participants who answered the questionnaires. Evaluation of the completed questionnaires revealed that five patients did not answer correctly, resulting in their exclusion, for a total of 222 Portuguese patient participants. ABRAFIBRO is a Brazilian institution that helps patients with FM, with a total of more than 6000 people diagnosed with the disease. With the help of this institution, 580 FM patients were contacted, and 253 patients answered the proposed questionnaires. Of these, only two participants did not fill out the questionnaire correctly, resulting in the exclusion of these patients from the study, for a total of 251 Brazilian participants.

### 2.3. Instruments

#### 2.3.1. Fibromyalgia-Related Fatigue

The questionnaire used to measure fibromyalgia-related fatigue levels was the Multidimensional Diary of Fatigue-Fibromyalgia-17 (MDF-Fibro-17), using the validated and translated version for the Portuguese and Brazilian populations [62]. The MDF-Fibro-17 includes 17 questions evaluating the various dimensions of fatigue. The questionnaire comprises five sections: global fatigue experience, consisting of 4 questions, for example, “How severe was your fatigue today?”; physical fatigue, consisting of 3 questions, for example, “How heavy did your body feel today?”; cognitive fatigue, consisting of 4 questions, for example, “How difficult was it to think clearly due to tiredness today?”; motivation, measured by 3 questions, such as, ”How difficult was it to motivate yourself to do something today?”; and the role impact, composed of 3 questions, for example, “Did you do something slower because you were feeling tired today?”. Participants evaluated fatigue components using a scale ranging from 0 (“not at all”) to 10 (“extremely”). Higher values indicate a higher level of fatigue. Previous studies support the validity, reliability and use of this questionnaire [62,63]. However, in the present study, only global fatigue experience, and physical and cognitive fatigue were used, as several previous studies [32,48] showed that these factors affect more of the psychological components in FM patients.

#### 2.3.2. Hospital Anxiety and Depression Scale (HADS)

To measure anxiety and depression levels, the Portuguese version of the Hospital Anxiety and Depression Scale was used [64]. This instrument consists of 14 questions, seven of which assess all aspects of anxiety and the other seven of which measure depression levels. This questionnaire employs a 4-point Likert scale (from 0 to 3); higher values represent higher levels of anxiety and depression. A study by Bjelland et al. [65] demonstrated that the HADS Portuguese version is well-structured and is reliable and precise in terms of measuring depressive and anxious conditions, as well as symptoms of anxiety and depression disorders.

#### 2.3.3. Self-Esteem

The Portuguese version of the Rosenberg Self-Esteem Scale [66] was used to assess self-esteem. Although the original version contains 10 items, for present study, we considered the 5 positively coded items (e.g., “On the whole, I am satisfied with myself”), as the negatively coded items can contaminate results [67]. The participants answered on a four-point Likert scale, with possible responses varying between 1 (“strongly agree”) and 4 (“strongly disagree”). The Rosenberg Self-Esteem Scale is widely used, and previous studies support its validity and reliability [68].

#### 2.3.4. Satisfaction with Life

The Portuguese version of the Satisfaction with Life Scale, [69] was used to measure global cognitive judgement the subjects’ life satisfaction. This scale has five questions (for example, “I’m happy with my life.”), and participants responded using a 5-point Likert scale, where 1 represents strongly disagree and 5 represents strongly agree. A study by Teixeira et al. [70] demonstrated the reliability and validity of this instrument.

#### 2.3.5. Physical Activity

To measure physical activity levels, we used the International Physical Activity Questionnaire (IPAQ) short form [71]. This questionnaire has seven questions, which are related to intense, moderate and walking activities performed in the last seven days before the completion of the questionnaire [72].

Another value obtained through the IPAQ is an estimate of energy expenditure derived from levels of physical activity [72]. In other words, the data are converted into MET min/week by calculating the minutes per week in each type of activity performed by the patient according to the specific metabolic equivalent [72,73].

### 2.4. Statistical Analysis

Descriptive statistics and bivariate correlation were calculated for variables under analysis. The hypothesized model was test with AMOS v.27 based on a two-step maximum likelihood approach, as suggested by Kline [74]. The first step was a confirmatory factor analysis (CFA) to analyze the psychometric properties of the model. In particular, internal consistency was determined via composite reliability according to Raykov [75], with 0.70 as a cutoff value [76]; convergent validity was assessed through average variance extracted (AVE) [76], considering values higher than or equal to 0.50 as an adequate value; and discriminant validity via square correlations between factors, considering correlation coefficients lower than the AVE for each construct adequate [77]. Then, a second step was performed based on structural equation modeling (SEM) to analyze the purposed associations. Standardized direct and indirect effects on the outcome variable were analyzed, considering coefficients significant if the 95% confidence intervals (CIs) were greater than zero [78]. Bootstrap resampling (1000 samples) considering a bias-corrected 95% CI was used to assess the significance of the direct and indirect effects. The adequacy of the models was evaluated using the following traditional incremental and absolute indices: comparative fit index (CFI), Tucker–Lewis index (TLI), standard root mean residual (SRMR) and root mean square error of approximation (RMSEA) with its confidence interval (CI: 90%); the following cutoff values were adopted: CFI and TLI ≥ 0.90, SRMR and RMSEA ≤ 0.8 [76,79,80].

#### Multigroup Analysis

An SEM multigroup analysis was performed according to the suggestion of Byrne [79] to test whether the proposed SEM model can be replicated in groups with different characteristics. Hence, the procedures recommended by Byrne [79] and Cheung and Rensvold [81], were followed. First, the SEM model should be adjusted to each group under analysis, and the following invariance types should be respected: unconstrained model, measurement weights, structural weights, structural covariances, structural residuals and measurement residuals. Invariance assumptions were verified through the differences in CFI (∆CFI), using <0.01 a cutoff value [79].

Full information robust maximum likelihood (FIML) was used to handle the small amount of missing data at the item level (missing at random = 2%), as proposed by Enders [82]. Then, we analyzed descriptive statistics and bivariate correlations. The values of skewness and kurtosis (between −2 and +2 and −7 and +7, respectively) revealed no deviations from univariate normality [76]. However, the normalized estimate of Mardia’s coefficient of multivariate kurtosis was greater than 5.0 in all samples under analysis. Consequently, Bollen–Stine bootstrap (2000 samples) was employed for subsequent analysis [83].

## 3. Results

### 3.1. Preliminary Analysis

The measurement model, including the factors of global fatigue experience, physical fatigue, cognitive fatigue, depression, anxiety, self-esteem, satisfaction with life and physical activity exhibited adequate fit to the data in all samples under analysis: general sample: χ^2^ = 979. 83 (364), BS-*p* < 0.001, CFI = 0.939, TLI = 0.929, RMSEA = 0.062 90% (0.057, 0.066) and SRMR = 0.043; Brazilian Sample: χ^2^ = 895. 83 (364), BS-*p* < 0.001, CFI = 0.922, TLI = 0.919, RMSEA = 0.073 90% (0.063, 0.078) and SRMR = 0.052; Portuguese Sample: χ^2^ = 921.64 (364), BS-*p* < 0.001, CFI = 0.932, TLI = 0.921, RMSEA = 0.061 90% (0.055, 0.067) and SRMR = 0.045. CFI and TLI were above and SRMR and RMSEA were below to the previously reported cutoff values.

### 3.2. Descriptive Statistics and Bivariate Correlations

Descriptive statistics showed that the participants presented scores above the midpoint for fibromyalgia-related fatigue, depression, anxiety and self-esteem and scores below the midpoint for satisfaction with life. The participants also showed a total amount of physical activity above 1000 Metz in the general sample. The same tendency was observed across the Portuguese and Brazilian samples, except for depression, for which the obtained values were below midpoint, and in terms of physical activity; the Portuguese sample seemed to be physically more active than the Brazilian patients (MET = 1721 vs. MET = 1334). With respect to correlation patterns in all samples analyzed, fibromyalgia-related fatigue was positively associated with depression and anxiety and negatively associated with self-esteem and satisfaction with life. In addition, fibromyalgia-related fatigue, particularly cognitive fatigue, showed a negative and significant association with physical activity, whereas global fatigue experience and physical fatigue did not present any association with physical activity. Depression and anxiety showed a negative and significant association with self-esteem, satisfaction with life and physical activity, whereas self-esteem and satisfaction with life were positively and significantly associated, as well as with physical activity, as shown in Table 1, Table 2 and Table 3.

### 3.3. Direct and Indirect Regression Paths

The results from the SEM analysis showed that the structural model provided an acceptable fit to the data in all samples: General sample: χ^2^ = 2254. 95 (382), BS-*p* < 0.001, CFI = 0.949, TLI = 0.938, RMSEA = 0.071 90% (0.053, 0.087) and SRMR = 0.072; Brazilian sample: χ^2^ = 1993.27 (382), BS-*p* < 0.001, CFI = 0.942, TLI = 0.929, RMSEA = 0.074 90% (0.057, 0.091) and SRMR = 0.071; and Portuguese sample: χ^2^ = 1883.37 (382), SRMR = 0.073, BS-*p* < 0.001, CFI = 0.9446, TLI = 0.932, RMSEA = 0.071 90% (0.052, 0.083) and SRMR = 0.070.

Table 4 shows the direct regression paths among studied variables in all samples. Specifically, the following associations were observed: (i) fibromyalgia-related fatigue showed a positive and significant association with anxiety and depression, (ii) depression and anxiety displayed a negative and significant association with self-esteem, (iii) self-esteem demonstrated a positive and significant association with satisfaction with life and (iv) satisfaction with life displayed a positive and significant association with physical activity.

The indirect regression paths, in all samples, showed that neither global fatigue experience nor physical fibromyalgia-related fatigue are indirectly associated with self-esteem, satisfaction with life and physical activity through depression and anxiety, self-esteem and satisfaction with life, respectively. However, fibromyalgia-related fatigue in terms of cognitive fatigue showed a positive indirect association with self-esteem and satisfaction with life and a positive indirect association with physical activity, with depression and anxiety, self-esteem and satisfaction with life as a possible mediator, respectively. In addition, depression and anxiety displayed a negative indirect association with satisfaction with life via self-esteem. Nevertheless, depression and anxiety showed a positive and negative indirect association with physical activity, with satisfaction with life as a mediator. Furthermore, self-esteem displayed a negative association with physical activity via satisfaction with life as possible mediator, as shown in Table 5.

### 3.4. Multigroup Analysis

The results of multigroup analysis (see Table 6) evidenced that the hypothesized SEM model was invariant between countries, as all the invariance criteria were attained, meaning that all factor loadings, structural paths, factor covariances, factor residual variances and measurement error variances were equivalent between countries (∆CFI < 0.01).

## 4. Discussion

### 4.1. Descriptive Statistics and Bivariate Correlations

The aim of this study was to analyze the associations across fibromyalgia-related fatigue, depression, anxiety, self-esteem satisfaction with life and physical activity. We obtained interesting findings that may add to the scarce literature on fibromyalgia-related fatigue. In addition, the stability of the proposed model was tested by means of cross-country (Brazil and Portugal) multigroup analyses. Initially, five hypotheses were proposed, which will be discussed according to the current literature.

With respect to the values of descriptive statistics, the composite reliability of each factor showed an adequate internal consistency for all studied variables in all samples under analysis, as the obtained values were higher than to 0.70. In addition, convergent validity was found to be acceptable (AVE ≥ 0.50), and discriminant validity was also adequate due to the square correlations between factors and AVE of each factor, as shown in Table 1, Table 2 and Table 3. In summary, these results support the preliminary conditions necessary to perform SEM analysis, as well as to analyze the direct and indirect effects among the studied variables.

The participants in this study presented scores above the midpoint for fatigue, depression, anxiety and self-esteem related to fibromyalgia and values below the midpoint for satisfaction with life; this scenario was verified both in the global sample and in the samples separated between Brazilian and Portuguese populations. These results were expected according to the current literature; however a factor of difference was that our patients demonstrated above-average values for self-esteem, in contrast to the results of other studies [30,58], which demonstrated that patients tend to have lower levels of self-esteem due to the adversity caused by FM.

### 4.2. Direct and Indirect Regression Paths

Regarding the first hypothesis, our results showed that fibromyalgia-related fatigue is positively associated with depression and anxiety and negatively associated with levels of self-esteem, life satisfaction and physical activity, confirming our expectations, thus demonstrating how fibromyalgia-related fatigue may be an important aspect of psychological aspects. Owing to these associations, it is likely that patients with higher levels of fatigue tend to have higher levels of depression and lower levels of self-esteem, satisfaction and physical activity. A study carried out by Kurtze et al. [84] corroborates these findings, revealing that when fatigue levels in patients with FM increased, anxiety and depression levels also increased, demonstrating a positive association between these components.

Our findings are in line with the results of several previous studies [37,38], which demonstrated that fatigue is negatively associated with life satisfaction and self-esteem. In their study, Galvez-Sanchez et al. [58] explained that this negative association might occur because patients with FM require a increased effort, with above-normal pain, to perform e tasks (such as daily activities and work tasks) due to their fatigue, thus causing a feeling of frustration and decreased expectation of self-efficacy, directly affecting the levels of self-esteem and satisfaction with life. This assumption seems to be confirmed by Johnson et al. [85], who it demonstrated that chronicity of Fibromyalgia-related fatigue ends up exposing the patient to extreme situations, which causes long-term psychological damage, thus affecting his/her self-confidence, self-esteem and satisfaction with life.

Another interesting finding of the present study is the negative and significant association between cognitive fatigue and physical activity levels, whereas the overall experience of fatigue and physical fatigue showed no association. Van As et al. [86] defined cognitive fatigue as a psychobiological state characterized by cognitive wasting, causing a series of negative feelings and leading to low levels of energy and positive affect in patients, resulting in decreased performance of physical activities. However, studies by Lukkahatai et al. [87] and Vincent et al. [88] showed that fatigue and all its domains are negatively and significantly associated with physical activity, demonstrating that there is a need for further in-depth studies to verify these associations between the domains of fatigue and physical activity.

In the case of the second hypothesis, we confirmed the negative association between the levels of depression and anxiety with self-esteem, thus confirming our expectations, suggesting that patients who present with higher levels of anxiety and depression tend to have lower levels of self-esteem. A study by Michalak et al. [89] corroborates the findings of this study, demonstrating a negative association between depression and anxiety with self-esteem.

This negative association between depression and anxiety with self-esteem can be explained by the fact that anxious and depressive conditions directly affect the lifestyle of FM patients, leading them to opt for social isolation and fostering a feeling of frustration and sadness, thus causing a decrease in self-esteem in these patients [26,27]. Sowislo et al. [90] performed a review of the associations between depression, anxiety and self-esteem and observed that low self-esteem is a consequence and not a causative factor because depressive episodes leave permanent scars on the self-concept, as in the anxiety state, it can generate a feeling of constant threat in patients and can reduce self-concept, thereby reducing self-esteem, demonstrating that depression and anxiety share high negative affectivity, i.e., stable disposition to non-specific distress and unpleasant mood, affecting levels of self-esteem.

In the present study, we also observed a positive association between the levels of self-esteem and life satisfaction. This shows that patients with higher levels of self-esteem tend to experience improved life satisfaction. Some studies [32,91] have shown that FM patients with low levels of self-esteem have reduced cognitive performance, especially in terms of attention, memory and planning skills, thus affecting their self-identity and their life satisfaction. Some studies [92,93] have shown that self-esteem acts as a strong predictor of life satisfaction because self-esteem is considered an important subjective construct based on personal perception and evaluation, involving not only the emotional aspect but also the performative aspect of functioning, which, once decreased, affects the levels of satisfaction in various areas of life (i.e., interpersonal relationships, cognitive and health changes, success and failure to perform tasks).

Our results further confirmed another hypothesis; levels of life satisfaction are positively associated with physical activity, which can demonstrate the need for health professionals seek ways for FM patients to feel more satisfied with life by increasing levels of physical activity. Previous studies [45,94] have shown that people with higher levels of life satisfaction have higher levels of physical activity and physical fitness. A possible explanation for this positive association is that physical activity, besides having a physiological impact, also affects the subjective wellbeing of the patient, thus improving psychological components, especially the levels of life satisfaction [45,47,48].

With respect to indirect effects, the overall experience of fatigue and physical fatigue showed no association with self-esteem, life satisfaction and physical activity, but cognitive fatigue showed an indirect positive association with self-esteem, life satisfaction and physical activity, with the amount of depression and anxiety, self-esteem and life satisfaction as possible mediators, respectively. In addition, depression and anxiety showed a negative indirect association with life satisfaction through self-esteem. However, depression and anxiety showed an indirect positive and negative association with physical activity, respectively, with life satisfaction as a mediator. It was also possible to verify that self-esteem showed a negative association with physical activity, with life satisfaction as a possible mediator. These analyses of the indirect regression pathway provided additional information supporting our hypotheses and identified some important new perceptions related FM and how fatigue relates to psychological components and physical activity levels.

The results of this study support the significant role FM as a complex health condition, whereby therapies, especially physical activity, should focus on improving subjective wellbeing because a according to some cross-sectional studies [94,95,96], FM patients with better physical fitness and higher levels of physical activity are associated with lower levels of depression, fatigue, anxiety and higher quality of life, life satisfaction and self-esteem.

### 4.3. Multigroup Analysis

The results of multigroup analysis showed that the model was structurally invariant across countries, confirming the similarity of the model across two investigated cultures. More specifically, the results demonstrate that the variables underlying the structural model were perceived in the same way by Brazilian and Portuguese FM patients and that the hypothesized relationships in the model could be interpreted in a similar way and with equivalent associations for all groups. Furthermore, these results are not only important because they support the relationships between fibromyalgia-related fatigue, including the experience of global fatigue, physical and cognitive fatigue, depression, anxiety, self-esteem, life satisfaction and physical activity, but also support the notion that the expected relationships between constructs may be generalizable to both Portugal and Brazil. Furthermore, this exposes the importance of these associations in PA of FM patients, reinforcing the suitability of this model for FM patients to increase their PA.

To the best of our knowledge, this is the first study to simultaneously examine the cross-cultural invariance between Portuguese and Brazilian populations considering all of these variables; however, our results disproved our hypothesis that there were cultural differences in terms of the associations between these variables, as suggested by previous literature [55,56,57,97,98] reporting differences in associations and perceptions of FM symptoms between cultures. Kuppens et al. [55] conducted a comparative study of FM patients from different cultures and found that Belgian patients demonstrated higher levels of fatigue and increased severity in psychological symptoms when compared to Dutch patients. This scenario was verified by Ruiz-Montero et al. [97], who found that Spanish patients showed higher values of associations between fatigue, anxiety and depression when compared to patients from Sweden, Belgium and the Netherlands. Our findings are in line with the results of a study by Clark et al. [57], who compared the perception and association of fatigue, anxiety, depression, difficulty in concentrating and pain symptoms of FM between European and Latin American patients and found that Latin American patients reported higher levels of these symptoms. Therefore, the results of this study reinforce the need for more in-depth studies to verify the influence of cultural aspects on the associations between fibromyalgia-related fatigue, anxiety, depression, self-esteem, satisfaction with life and physical activity.

### 4.4. Practical Implications and Limitations

The number of scientific publications on FM, especially in the area of therapies for symptom control, has increased in recent years, as it has been verified that interventions, among which physical activity stands out, have achieved positive results in treating the symptoms of FM patients. The results of this study offer a significant contribution to the literature with respect to the association between fibromyalgia-related fatigue, psychological components and physical activity, helping to fill a gap in the literature, as it is the first study to verify the association of all these variables simultaneously, which demonstrates how FM is a delicate and complex health condition that must be further studied and explored to ensure a better quality of life for FM patients. The uniqueness of this study further reinforces the importance of assessing fatigue in patients with FM, as it is one of the main barriers to physical activity and directly affects the levels of anxiety, depression, life satisfaction and self-esteem, independent of culture.

Although the present study contributes new knowledge and data on the associations between fibromyalgia-related fatigue, depression, anxiety, self-esteem, life satisfaction, and physical activity, it is subject to some limitations that should be addressed. The present study was conducted with a cross-sectional design; further studies should be conducted using different approaches (i.e., longitudinal or experimental) to verify the associations between the variables investigated in this study. In addition, physical activity was evaluated through a subjective questionnaire, so the actual intensity of each activity reported by the patients was not necessarily considered due to a lack of supervisors to verify the conditions under which such activities were performed. In the present study, only female patients answered the proposed questionnaires, preventing generalization of the findings to male patients with FM. Future studies should include male patients for comparison with the results presented herein. A further limitation of this study is that we did not verify whether the patients used any type of medication to control FM symptoms. Therefore, futures studies should make an effort to collect data on this indicator for use as a possible moderator across studied variables.

## 5. Conclusions

One of the main strengths of the current study is the considerable sample size of patients with FM, from which it was possible to verify the relationships between the studied variables. Another positive aspect of our study is the use of psychometrically valid and reliable measures to evaluate of the proposed variables. The use of these instruments allowed us to make more reliable comparisons between studies. Another strong point is the uniqueness of this study, as it is the first study to analyze and compare the proposed variables simultaneously, in addition to a comparison between two distinct cultures.

In general, our results showed significant associations between fibromyalgia-related fatigue, depression, anxiety, self-esteem, life satisfaction and physical activity, with no difference in perception and association between two distinct cultures. Thus, patients with FM who have higher levels of fatigue, especially in the domain of cognitive fatigue, tend to have higher levels of anxiety and depression, causing a possible decrease in levels of self-esteem, life satisfaction and physical activity. In addition, it seems that fibromyalgia-related fatigue, depression and anxiety, self-esteem, satisfaction of with life and physical activity follow a similar trend, independent of the patient’s culture.

## Figures and Tables

**Table 1 medicina-58-01097-t001:** Descriptive statistics, bivariate correlations, average variance extracted values and composite reliability coefficients in the general sample.

Variable	M	SD	1	2	3	4	5	6	7	8	AVE	CR
1. GFE	7.66	1.71	1	-	-	-	-	-	-	-	0.83	0.95
2. PF	7.98	1.69	0.87 **	1	-	-	-	-	-	-	0.87	0.95
3. CF	7.44	1.88	0.78 **	0.77 **	1	-	-	-	-	-	0.79	0.92
4. Depression	1.52	0.67	0.36 **	0.36 **	0.39 **	1	-	-	-	-	0.51	0.80
5. Anxiety	2.01	0.68	0.35 **	0.37 **	0.38 **	0.67 **	1	-	-	-	0.50	0.79
6. Self-Esteem	3.01	0.92	−0.14 *	−0.10 **	−0.12 **	−0.55 **	−0.36 **	1	-	-	0.53	0.82
7. SWL	2.40	0.87	−0.34 **	−0.29 **	−0.38 **	−0.69 **	−0.56 **	0.45 **	1	-	0.87	0.87
8. PA	1527	401.83	0.02	0.01	−0.24 **	−0.10 **	−0.17 **	−0.16 **	−0.18 **	1	-	-

M = mean; SD = standard deviation; GFE = global fatigue experience; PF = physical fatigue; CF = cognitive fatigue; SWL = satisfaction with life; PA = physical activity; AVE = average variance extracted; CR = composite reliability. * *p* < 0.05; ** *p* < 0.01.

**Table 2 medicina-58-01097-t002:** Descriptive statistics, bivariate correlations, average variance extracted values and composite reliability coefficients in the Brazilian sample.

Variable	M	SD	1	2	3	4	5	6	7	8	AVE	CR
1. GFE	7.67	1.74	1	-	-	-	-	-	-		0.83	0.94
2. PF	7.98	1.73	0.88 **	1	-	-	-	-	-		0.86	0.95
3. CF	7.45	1.91	0.79 **	0.77 **	1	-	-	-	-		0.79	0.93
4. Depression	1.41	0.68	0.36 **	0.37 **	0.41 **	1	-	-	-		0.50	0.79
5. Anxiety	2.02	0.69	0.35 **	−0.11 **	0.39 **	0.69 **	1	-	-		0.50	0.79
6. Self-Esteem	3.02	0.91	−0.16 *	−0.31 **	−0.13 **	−0.56 **	−0.38 **	1	-		0.54	0.81
7. SWL	2.41	0.87	−0.33 **	−0.32 **	−0.39 **	−0.71 **	−0.55 **	0.48 **	1		0.88	0.87
8. PA	1334	368.12	0.04	0.03	−0.26 **	−0.11 **	−0.18 **	−0.18 **	−0.21 **		-	-

M = mean; SD = standard deviation; GFE = global fatigue experience; PF = physical fatigue; CF = cognitive fatigue; SWL = satisfaction with life; PA = physical activity; AVE = average variance extracted; CR = composite reliability. * *p* < 0.05; ** *p* < 0.01.

**Table 3 medicina-58-01097-t003:** Descriptive statistics, bivariate correlations, average variance extracted values and composite reliability coefficients in the Portuguese sample.

Variable	M	SD	1	2	3	4	5	6	7	8	AVE	CR
1. GFE	7.67	1.68	1	-	-	-	-	-	-		0.84	0.95
2. PF	7.98	1.66	0.89 **	1	-	-	-	-	-		0.86	0.95
3. CF	7.42	1.85	0.79 **	0.78 **	1	-	-	-	-		0.80	0.94
4. Depression	1.39	0.67	0.37 **	0.38 **	0.42 **	1	-	-	-		0.51	0.80
5. Anxiety	2.10	0.67	0.35 **	−0.13 **	0.40 **	0.70 **	1	-	-		0.51	0.79
6. Self-Esteem	3.00	0.93	−0.17 *	−0.32 **	−0.14 **	−0.57 **	−0.39 **	1	-		0.54	0.81
7. SWL	2.40	0.85	−0.34 **	−0.34 **	−0.41 **	−0.70 **	−0.56 **	0.49 **	1		0.89	0.88
8. PA	1721	575.04	0.06	0.04	−0.27 **	−0.12 **	−0.19 **	−0.18 **	−0.23 **		-	-

M = mean; SD = standard deviation; GFE = global fatigue experience; PF = physical fatigue; CF = cognitive fatigue; SWL = satisfaction with life; PA = physical activity; AVE = average variance extracted; CR = composite reliability. * *p* < 0.05; ** *p* < 0.01.

**Table 4 medicina-58-01097-t004:** Direct regression paths.

Path	Effect	CI 95%	*p*
GFE→Depression	0.14	[0.073, 0.318]	0.003
GFE→Anxiety	0.18	[0.133, 0.287]	0.002
PF→Depression	0.19	[0.132, 0.325]	0.004
PF→Anxiety	0.16	[0.142, 0.268]	0.003
CF→Depression	0.25	[0.066, 0.409]	0.001
CF→Anxiety	0.23	[0.035, 0.387]	0.001
Anxiety→Self-Esteem	−0.29	[−0.661, −0.376]	0.001
Depression→Self-Esteem	−0.53	[−0.656, −0.386]	0.001
Self-Esteem→SWL	0.51	[0.395, 0.612]	0.001
SWL→PA	0.27	[0.315, 0.566]	0.001
Brazilian Sample			
GFE→Depression	0.16	[0.071, 0.388]	0.025
GFE→Anxiety	0.20	[0.022, 0.298]	0.002
PF→Depression	0.18	[0.135, 0.329]	0.003
PF→Anxiety	0.19	[0.157, 0.279]	0.002
CF→Depression	0.28	[0.086, 0.433]	0.001
CF→Anxiety	0.25	[0.045, 0.3418]	0.001
Anxiety→Self-Esteem	−0.31	[−0.768, −0.299]	0.001
Depression→Self-Esteem	−0.55	[−0.686, −0.359]	0.001
Self-Esteem→SWL	0.52	[0.399, 0.652]	0.001
SWL→PA	0.29	[0.352, 0.598]	0.001
Portuguese Sample			
GFE→Depression	0.17	[0.079, 0.396]	0.003
GFE→Anxiety	0.21	[0.196, 0.315]	0.002
PF→Depression	0.18	[0.128, 0.314]	0.003
PF→Anxiety	0.19	[0.192, 0.303]	0.002
CF→Depression	0.28	[0.101, 0.514]	0.001
CF→Anxiety	0.27	[0.034, 0.413]	0.001
Anxiety→Self-Esteem	−0.28	[−0.651, −0.356]	0.001
Depression→Self-Esteem	−0.54	[−0.676, −0.317]	0.001
Self-Esteem→SWL	0.50	[0.399, 0.599]	0.001
SWL→PA	0.30	[0.403, 0.678]	0.001

GFE = global fatigue experience; PF = physical fatigue; CF = cognitive fatigue; SWL = satisfaction with life; PA = physical activity; CI 95% = confidence interval at 95%; *p* = level of significance.

**Table 5 medicina-58-01097-t005:** Indirect regression paths.

Path	Effect	CI 95%	*p*
General Sample			
GFE→Self-Esteem	−0.07	[−0.203, 0.047]	0.230
GFE→SWL	−0.04	[−0.111, 0.024]	0.221
GFE→PA	0.004	[−0.001, 0.018]	0.148
PF→Self-Esteem	−0.06	[−0.201, 0.072]	0.340
PF→SWL	−0.03	[−0.111, 0.038]	0.323
PF→PA	0.005	[−0.003, 0.017]	0.216
CF→Self-Esteem	−0.15	[−0.271, −0.044]	0.005
CF→SWL	−0.08	[−0.151, −0.002]	0.005
CF→PA	0.12	[0.056, 0.189]	0.011
Anxiety→SWL	−0.05	[−0.115, −0.012]	0.032
Depression→SWL	−0.27	[−0.386, −0.165]	0.005
Anxiety→PA	−0.006	[−0.001, 0.019]	0.099
Depression→PA	0.03	[0.009, 0.061]	0.015
Self-Esteem→PA	−0.06	[−0.110, −0.013]	0.020
Brazilian Sample			
GFE→Self-Esteem	−0.06	[−0.193, 0.042]	0.267
GFE→SWL	−0.03	[−0.012, 0.018]	0.334
GFE→PA	0.006	[−0.003, 0.021]	0.251
PF→Self-Esteem	−0.06	[−0.201, 0.072]	0.340
PF→SWL	−0.05	[−0.211, 0.048]	0.381
PF→PA	0.004	[−0.002, 0.012]	0.316
CF→Self-Esteem	−0.18	[−0.381, −0.074]	0.004
CF→SWL	−0.09	[−0.162, −0.005]	0.005
CF→PA	0.12	[0.077, 0.201]	0.009
Anxiety→SWL	−0.07	[−0.134, −0.034]	0.029
Depression→SWL	−0.28	[−0.401, −0.179]	0.004
Anxiety→PA	−0.004	[−0.001, 0.014]	0.114
Depression→PA	0.06	[0.010, 0.091]	0.012
Self-Esteem→PA	−0.08	[−0.123, −0.015]	0.019
Portuguese Sample			
GFE→Self-Esteem	−0.09	[−0.214, 0.057]	0.211
GFE→SWL	−0.05	[−0.124, 0.057]	0.221
GFE→PA	0.002	[−0.001, 0.010]	0.377
PF→Self-Esteem	−0.08	[−0.390, 0.096]	0.290
PF→SWL	−0.04	[−0.123, 0.047]	0.297
PF→PA	0.004	[−0.002, 0.012]	0.236
CF→Self-Esteem	−0.17	[−0.292, −0.057]	0.005
CF→SWL	−0.10	[−0.167, −0.006]	0.004
CF→PA	0.14	[0.066, 0.212]	0.008
Anxiety→SWL	−0.07	[−0.127, −0.023]	0.028
Depression→SWL	−0.29	[−0.395, −0.178]	0.004
Anxiety→PA	−0.004	[−0.001, 0.014]	0.100
Depression→PA	0.05	[0.091, 0.067]	0.020
Self-Esteem→PA	−0.07	[−0.1117, −0.012]	0.023

GFE = global fatigue experience; PF = physical fatigue; CF = cognitive fatigue; SWL = satisfaction with life; PA = physical activity; CI 95% = confidence interval at 95%; *p* = level of significance.

**Table 6 medicina-58-01097-t006:** Goodness-of-fit indices for the invariance of the structural model between countries.

Model	χ^2^	df	∆ χ^2^	∆df	*p*	CFI	∆CFI
Brazilian vs. Portuguese							
UM	2978.084	1170	-	-	-	0.933	-
MW	2980.674	1199	2.580	29	0.026	0.931	0.002
SW	2981.360	1208	3.276	38	0.054	0.930	0.003
SR	2983.118	1215	5.030	45	<0.001	0.929	0.004
MR	2989.010	1251	10.926	81	<0.001	0.921	0.008

χ² = chi-square; ∆χ² = differences in value of chi-square; ∆df = differences in degrees of freedom; *p* = level of significance; CFI = comparative fit index; ∆CFI = differences in the value of the comparative fit index; UM: unconstrained model; MW: measurement weight; SW: structural weight; SR: structural residual; MR: measurement residual.

## Data Availability

Due to issues related to participant consent, data will not be shared publicly. Interested researchers may contact the corresponding author (diogo.monteiro@ipleiria.pt).

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
