# Peer review of "Understanding the Associations across Fibromyalgia-Related Fatigue, Depression, Anxiety, Self-Esteem Satisfaction with Life and Physical Activity in Portuguese and Brazilian Patients: A Structural Equation Modeling Analysis"

_medicina, 2022, doi:10.3390/medicina58081097_

Round 1

Reviewer 1 Report

Review the document.

Author Response

Reviewer 1:

Dear Authors, Interesting work, well-structured in content and form. Important contributions are made on fibromyalgia and related variables such as behavioral and modifiable lifestyles. In general terms, some corrections in the results and discussion section are recommended with a view to improving the manuscript.

R: We appreciate your time and effort on the review the manuscript.

The summary is very long, maximum 200 words

R: The summary has been rewritten.

We could say that, in the total sample, when the PA of the questionnaire increases by one unit, the SWL increases by 0.27 units. This can be interpreted for all associations. What do the authors think? (Line 393 and Table 2)

R: We thank the reviewer for his/her comment. Regarding the question made by reviewer, the values reported in table 2 are related to the associations across studied variables in terms of variance explained by each variable. This is the interpretation in terms of Structural Equation Modeling analysis (Byrne, 2016; Hair et al., 2019.

The indirect association. Do the authors say that each association was adjusted for confounding variables? For example, self-esteem with PA, adjusted for SWL. They need to clear this up. Then, it could be interpreted as: when PA increases by one unit (in IPAQ), self-esteem decreases by -0.06 units when adjusting for SWL. Do the authors agree? (Line 411 and Table 3).

R: We thank the reviewer for his/her comment. The indirect effects reported in table 3, are the indirect effects observed across studied variables. In other words, depression presented a negative indirect association with SWL through self-esteem. It means that self-esteem could act as a buffer in relationship between depression and SWL. The remain indirect effects are interpreted in same way as recommended by several authors (e.g., Byrne, 2016; Hair et al., 2019).

In Table 2 and 3, does the term “effect” refer to the beta coefficient?

R: Yes, the term "effect" refers to the Beta coefficient. 

“cross-sectional design”. As a complement, a cause-effect dynamic between the variables cannot be assumed (line 680).

R: The reviewer is right. The sentence was rephrased.

This is not clear (line 684-685), since it has been mentioned that it is possible to generalize the results according to those stated in lines 642-643. Although a sample calculation was made based on an equation (line 216), this (line 684-685) confuses the reader.

R: The reviewer is correct. The sentence has been removed.

This is not a limitation since it was implicit in the objective of the study (line 689).

R: The reviewer is correct. So the part where we suggested it was a limitation of the present study was removed, but we left as a recommendation for a future study the use of male patients.

Are there no strengths in the study? Authors must include strengths (line 695)

R: The reviewer is correct, as indeed it is positive points in this study that were, which were included.

Reviewer 2 Report

In addition to diffuse pain in skeletal muscles and the physical finding of multiple tender points, most patients with fibromyalgia report fatigue, muscle stiffness, pain after physical exertion, and sleep abnormalities. Congratulations to the authors for studying these other aspects of the very complex and painful fibromyalgia syndrome. 

The text still needs to be better written to be published, it is too long and complex to read.​​​​​​​ It needs to be more concise and direct to the reader.

Abstract

The Results and Conclusion part must be redone according to the observations below and the text of the article. Take the time to polish the Abstract. Make a good first impression with a well-written abstract. Clearly, describe the problem in the first sentence of the abstract. Describe your primary conclusions better.

Tender points

The diagnostic criteria for fibromyalgia have since changed, and tender points are no longer used as a diagnostic tool. Fibromyalgia is currently diagnosed through a series of laboratory tests that rule out other causes of widespread pain, fatigue, and sleep disturbance. These new techniques have identified many more people with fibromyalgia who didn’t meet the tender points criteria. Over the years, many people with fibromyalgia have gone undiagnosed because they didn’t have tenderness in 11 spots at the time of their physical exam. As of 1990, tender points began to be used as one of the diagnostic criteria for fibromyalgia, which is used by the American College of Rheumatology (ACR). However, in 2010, the ACR released the Preliminary Diagnostic Criteria for Fibromyalgia. This document established additional criteria for its diagnosis, using the Generalized Pain Index (GDI) and the Symptom Severity Scale (EGS). They were also created for use at the primary level and do not require the physical examination of the tender points (due to the finding of little use in medical practice). The authors used SSS and WPI criteria (lines 64–75) that predate the ACR 2010 update.

Update this in the introduction, abstract especially as the authors cite the article with updated criteria in the reference (Wolfe et al., 2010).

1.4 Interesting paragraph comparing Brazilian and Portuguese populations regarding the characteristics of fibromyalgia, this could be better explored in the introduction to give more sense of the context and objective of this article with authors from both countries.

Write a concise, focused Introduction. It is too long and contains too much information, too many references, and not enough punch. You can improve your Introduction by explaining why your research question is important, interesting, or controversial, but exclude paragraphs of information that can be found in a textbook. Although you may have conducted an exhaustive review of the literature, include only the most relevant and significant points in your Introduction.   

Write a strong introductory paragraph, and go right to the essence of the essence of the argument to “hook” the reader. Start the Introduction with a general, yet concise, description of the problem that your paper will address. In the next sentences, reference previous work that supports your assessment of the problem. Be more concise. Early in the Introduction define the primary subject of your paper.

Method

It is well described and seems reproducible.

Results

Start the Results section with the major positive findings. Report results in the Medicine journal format. Do not mix references in the result that should be explained in the Method section (95, 89, 96). The Results section rarely requires references. A sentence that requires a reference probably belongs to another section. Include only the results in the Results section. Focus on your hypothesis, and move any interpretations of the results to the Discussion (p.ex. line 352–358). Separate the Results and the Discussion – completely.

The description in Table 1 is crucial, it is the essence of this study. Design reader-friendly tables. It better describe the legend of all Tables so that the reader does not need to read the entire result to understand.  It must be simple and self-explanatory.  The Table legends are incomplete and difficult for the reader to understand what it is all about just by looking at the Table. It needs to add a more explanatory text in the subtitles.

Discussion

Star the Discussion with your most important point. Your discussion should start with one sentence that clearly shows that your paper contains new information. Discuss the financial implications of your findings if possible. Consider alternative explanations for your results. In the Discussion, consider the opposing viewpoint by taking the devil´s advocate position. Identify the strengths and note the weakness of your study, explore more this.   

What do the authors recommend? What are the clinical implications of your findings?

Conclusion

Conclude the Discussion with a “bolt of lightning.” The Introduction must begin with thunder and Conclusion ends with lightning.

Ref.:

Wolfe F, Clauw DJ, Fitzcharles M, Goldenberg DL, Katz RS, Mease P et al. The American College of Rheumatology Preliminary Diagnostic Criteria for Fibromyalgia and Measurement of Symptom Severity. Arthritis Care Res. 2010; 62(5):600–10.

Author Response

Reviewer 2:

In addition to diffuse pain in skeletal muscles and the physical finding of multiple tender points, most patients with fibromyalgia report fatigue, muscle stiffness, pain after physical exertion, and sleep abnormalities. Congratulations to the authors for studying these other aspects of the very complex and painful fibromyalgia syndrome.

R: We thank the reviewer for his/her comment.

The text still needs to be better written to be published, it is too long and complex to read.​​​​​​​ It needs to be more concise and direct to the reader.

R: We thank the reviewer for the time spent reviewing our study and for the comments made for its improvement.

Abstract

The Results and Conclusion part must be redone according to the observations below and the text of the article. Take the time to polish the Abstract. Make a good first impression with a well-written abstract. Clearly, describe the problem in the first sentence of the abstract. Describe your primary conclusions better.

R: We thank the reviewer for his comment, and the abstract has been rewritten according to the changes made.

Tender points

The diagnostic criteria for fibromyalgia have since changed, and tender points are no longer used as a diagnostic tool. Fibromyalgia is currently diagnosed through a series of laboratory tests that rule out other causes of widespread pain, fatigue, and sleep disturbance. These new techniques have identified many more people with fibromyalgia who didn’t meet the tender points criteria. Over the years, many people with fibromyalgia have gone undiagnosed because they didn’t have tenderness in 11 spots at the time of their physical exam. As of 1990, tender points began to be used as one of the diagnostic criteria for fibromyalgia, which is used by the American College of Rheumatology (ACR). However, in 2010, the ACR released the Preliminary Diagnostic Criteria for Fibromyalgia. This document established additional criteria for its diagnosis, using the Generalized Pain Index (GDI) and the Symptom Severity Scale (EGS). They were also created for use at the primary level and do not require the physical examination of the tender points (due to the finding of little use in medical practice). The authors used SSS and WPI criteria (lines 64–75) that predate the ACR 2010 update. Update this in the introduction, abstract especially as the authors cite the article with updated criteria in the reference (Wolfe et al., 2010).

R: The reviewer is correct. The use of tender points in the diagnosis of FM has been removed, and the text has been revised to update the current diagnostic criteria.

Write a concise, focused Introduction. It is too long and contains too much information, too many references, and not enough punch. You can improve your Introduction by explaining why your research question is important, interesting, or controversial, but exclude paragraphs of information that can be found in a textbook. Although you may have conducted an exhaustive review of the literature, include only the most relevant and significant points in your Introduction.

R: We thank the reviewer for his/her comment. The reviewer is right, and the introduction was redesigned.

Write a strong introductory paragraph and go right to the essence of the essence of the argument to “hook” the reader. Start the Introduction with a general, yet concise, description of the problem that your paper will address. In the next sentences, reference previous work that supports your assessment of the problem. Be more concise. Early in the Introduction define the primary subject of your paper.

R: We thank the reviewer for his/her comment. Indeed, the reviewer is right. Therefore, a paragraph was added in the introduction presenting the problematic of the study.

Method

It is well described and seems reproducible.

R: We appreciate the reviewer commentary.

Results

Start the Results section with the major positive findings. Report results in the Medicine journal format. Do not mix references in the result that should be explained in the Method section (95, 89, 96). The Results section rarely requires references. A sentence that requires a reference probably belongs to another section. Include only the results in the Results section. Focus on your hypothesis and move any interpretations of the results to the Discussion (p.ex. line 352–358). Separate the Results and the Discussion – completely.

R: We thank the reviewer for his/her comment. The references in Results section was moved to the methods section, and the interpretations was moved to the discussion section.

The description in Table 1 is crucial, it is the essence of this study. Design reader-friendly tables. It better describe the legend of all Tables so that the reader does not need to read the entire result to understand.  It must be simple and self-explanatory.  The Table legends are incomplete and difficult for the reader to understand what it is all about just by looking at the Table. It needs to add a more explanatory text in the subtitles.

R: We thank the reviewer for his/her comment. The table 1 was divided into three tables (one of them of each sample). Therefore, we think that in this way the tables are more friendly to the readers.

Discussion

Star the Discussion with your most important point. Your discussion should start with one sentence that clearly shows that your paper contains new information. Discuss the financial implications of your findings if possible. Consider alternative explanations for your results. In the Discussion, consider the opposing viewpoint by taking the devil´s advocate position. Identify the strengths and note the weakness of your study, explore more this.  

R: We thank the reviewer for his/her comment. The discussion has been readjusted, and a topic on the strengths of the study has been added.

What do the authors recommend? What are the clinical implications of your findings?

R: We thank the reviewer for his/her comment. We recommend further studies on fibromyalgia-related fatigue to better understand and control it, thus ensuring an improvement in the quality of life of patients with FM. In addition, the clinical implications topic has been revised.

Round 2

Reviewer 2 Report

Congratulations to all the authors, it was much more pleasant to read with the corrections that were indicated. It changed from water to a fine wine.